# Regional Features of the Arctic Sea Ice Area Changes in 2000–2019 versus 1979–1999 Periods

**Tatiana A. Matveeva** [1,2,*] **and Vladimir A. Semenov** [1,2]

1 Institute of Geography, Russian Academy of Science, Staromonetny Lane, 29, 119017 Moscow, Russia
2 A.M. Obukhov Institute of Atmospheric Physics, Russian Academy of Sciences, Pyzhyovskiy Lane, 119017 Moscow, Russia
* Correspondence: matveeva.tatiana@igras.ru

**Abstract:** One of the most striking manifestations of ongoing climate change is a rapid shrinking of the Arctic sea ice area (SIA). An important feature of the observed SIA loss is a nonlinear rate of a decline with an accelerated decrease in the 2000–2019 period relative to a more gradual decline in 1979–1999. In this study, we perform a quantitative assessment and comparison of the spatial-temporal SIA changes during these two periods. It was found that winter Arctic SIA loss is primarily associated with changes in the Barents Sea, where the SIA decline in 2000–2019 has accelerated more than three-fold in comparison with 1979–1999. In summer and autumn, rates of SIA decline in 2000–2019 increased most strongly in the Kara, Beaufort Seas, the Northwestern Passage, and inner Arctic Ocean. The amplitude of the SIA seasonal cycle has also increased in 2000–2019 in comparison with the earlier period, with the largest changes in the inner Arctic Ocean, the Kara, Laptev, East Siberian and Beaufort Seas in summer and in the Barents Sea in winter. The results may reflect a transition to a new dynamic state in the recent two decades with the triggering of positive feedbacks in the Arctic climate system.

**Keywords:** Arctic sea ice; Arctic climate; climate change; ERA5



## 1. Introduction

One of the most drastic manifestations of ongoing climate change is the rapid shrinking of the Arctic sea ice cover over the last decades (e.g., [1]). The sea ice plays an important role in the Arctic climate system and beyond. It takes part in a number of important climate feedbacks, modulates energy fluxes at the ocean-atmosphere interface, and interacts with atmospheric and oceanic circulation (e.g., [2–4]). The sea ice is also a crucial element for the Arctic biodiversity [5,6], and economic and social development. In particular, the Arctic sea ice decline facilitates exploration of resources and usage of marine transportation routes [7–10].

Since the beginning of satellite observation era in 1979, the total September Arctic sea ice area (SIA) (climatological minimum) has been declining on average by about 11% per decade [11,12]. Despite the SIA decline in summer being more than three times faster in relative percentage terms in comparison with the winter decline of 3%, the difference in absolute trend values is not so drastic due to the greater SIA in winter. The summer SIA absolute trend is only twice as fast as the winter one.

As sea ice melts, surface albedo decreases and the upper ocean absorbs more solar radiation that provokes a stronger melt (co-called ice–albedo feedback, e.g., [13,14]). This positive feedback is one reason for the Arctic amplification of the global warming [15], especially over the Arctic Ocean.

Importantly, the decrease has not been uniform over time and has accelerated to almost twice the previous rate since 2000 (e.g., [16,17]). Such rates of decrease imply a complete disappearance of the sea ice already by the 2030s [18].

The changes in the Arctic sea ice are manifested in several important characteristics including specific temporal and spatial patterns, timings of ice onset and removal, duration of the melt season [13,19], the thinning of first-year sea ice and decreases in multi-year ice area (e.g., [4,20]). The observed Arctic SIA decline is reasonably well simulated by modern climate models under scenarios of anthropogenic climate forcing, although the recent accelerated trends are on average somewhat underestimated [1,13]. This implies an important role of the external anthropogenic forcing in driving the observed Arctic sea ice changes, but also suggests a contribution from internal climate variability [21], which is highly non-uniform [22]. Natural variability causes variations in the oceanic and atmospheric heat inflows to the Arctic from the North Pacific [23] and the North Atlantic [4,24]. A long-term internal variation enhanced by positive feedbacks could explain a strong SIA decline during the Early Twentieth Century Warming indicated by observations, reconstructions and model simulations [25–28].

Sea ice loss features are regionally dependent, with different forcing factors and processes involved in different regions [22]. For example, [29] demonstrated an increasing role of ocean variability in the Arctic SIA reduction, in particular, the major role of the Atlantic inflow in the sea ice variations in the Barents, Kara, Laptev and East Siberian Seas [30,31]. This role will further increase in the future [32]. The important contribution of the internal multidecadal variability in the North Atlantic to the Arctic sea ice changes has been indicated by several studies (e.g., [33,34]). Furthermore, some recent studies also suggest an important contribution of the Pacific Ocean to the recent sea ice loss [35,36].

With the accelerated Arctic sea ice loss in the 21st century accompanied by cooling over North Eurasia, the interest in the links between sea ice and atmospheric circulation has resulted in an avalanche of studies (e.g., [37–42]).

The SIA changes impact atmospheric circulation with the response being dependent on the region and the magnitude of the sea ice loss (e.g., [3,43,44]). Atmospheric circulation in turn also affects sea ice variations, in particular, on interannual and decadal time scales. Large scale atmospheric variability modes including the Arctic Oscillation, Arctic Dipole, and Ural Blocking have been implicated in the summer SIA minima in 2007 and 2012, contribute to winter SIA decline, and may explain the absence of new summer SIA record minima since 2012 [45–47]. We note, however, that some studies attribute only a small fraction of the summer SIA decline in the recent decades to atmospheric circulation changes [48].

The changes of cloudiness may also play an important role in the variability and trends in the Arctic sea ice. It has been demonstrated that increasing cloud cover could have contributed significantly to the recent and historical sea ice changes [49–51]. This impact may increase as thinner ice, which has been occupying larger areas and replacing thicker ice in vast regions, is more sensitive to the changes of cloud cover [52].

One of the important features of the observed Arctic SIA loss is the nonlinear rate of the decline with the accelerated decrease in the beginning of the 21st century for the total Arctic SIA relative to the more gradual decline in previous decades, as well as its regional dependence. This implies an existence of tipping points in the sea ice dynamics, thresholds where the related processes qualitatively alter [53]. The issue of the sea ice tipping points remains controversial. Such tipping points suggest that the analysis of the sea ice changes should be performed for the periods before and after the point is reached [54,55]. The exact change point in the SIA time series is hard to define precisely, as it depends on both the season and region under consideration. There are a variety of studies dealing with statistical approaches to detect such points (e.g., [56,57]). Applying one of the methods [58] results in years 2004 and 2005 for March and September, respectively, for the total Arctic SIA, and in a range of years from 1996 to 2007 for different Arctic Seas. Considering the length of the analyzed SIA record spanning from 1979 to 2019 and the above mentioned uncertainties in defining change points, we choose to divide the period into two almost equal size sub-periods, 1979–1999 and 2000–2019, as a reasonable compromise.

Given the outlined spatial inhomogeneity of the Arctic sea ice changes and non-linearity of the changes during the last four decades, it is important to assess SIA changes

in the different Arctic seas for different periods. Several studies estimated changes of SIA in individual Arctic seas [12,59,60], but they did not focus on the difference in rates of SIA decline in different periods. The major questions we address in this study are the following: what are the regional features of the spatial-temporal SIA changes, and how do they differ during these two periods? Studying these issues is important for understanding the modern transformation of the Arctic climate system.

## 2. Materials and Methods

In contrast to many studies on Arctic sea ice, where sea ice extent (defined as a grid cell area with a sea ice concentration greater than 15%) is analyzed, we use sea ice area (SIA) as a measure characteristic of the sea ice coverage (i.e., integrated sea ice concentration, SIC, values without any threshold). The use of sea ice extent reduces errors associated with the uncertainty of observational data. However, a significant disadvantage of sea ice extent is the loss of information on SIC within a given area (where SIC is greater than 15% but less than 100%) because this SIC range contains significant variations which strongly affect turbulent heat fluxes at the ocean-atmosphere boundary (especially in winter) [1]).

The main modern data source of SIC is satellite microwave radiometers, which retrieve SIC from brightness temperature data using different algorithms. It was revealed by [61] that different algorithms for SIC retrieval result in noticeable differences in both absolute values and trends of estimated sea ice area and extent. For example, for the annual mean Arctic SIA, the difference among algorithms reached $1.3 \times 10^6$ km$^2$. It could be caused by different choice of the microwave channels, weather filters and other factors [62].

We use data on SIC from ERA5 Reanalysis [63], which includes passive microwave satellite data reprocessed by Ocean and Sea Ice Satellite Application Facility (OSI-SAF). OSI-SAF uses the combination of two SIC algorithms, called Bootstrap frequency mode [62] and Bristol [64], with different weights for different concentrations. The main advantage of OSI-SAF processed data over passive microwave satellite data is the use of numerical weather prediction data for correction of initial data on brightness temperatures prior to calculating SIC, which results in more reliable SIC estimates.

We compared the passive microwave satellite data NOAA/NSIDC Version 3 [65], ERA5 [63] and the Hadley Centre Sea Ice and Sea Surface Temperature data set Version 1 (HadISST1) [66]. Since the true values are basically unknown, and the data from different source show agreement in direction of SIA changes, we selected the ERA5 data due to the use of numerical weather prediction data for initial guess corrections, which is suggested to be a more advanced method.

The 1979–2019 period of continuous satellite observations was used for the analysis. Within this period, the total northern hemisphere (NH) sea ice cover has been changing with visibly different rates (Figure 1a). Therefore, two periods of almost the same length representing (relatively) slow and fast SIA declines were chosen for detailed consideration: 1979–1999 (SIA trends in March and September are $-145 \times 10^3$ km$^2$ per decade and $-411 \times 10^3$ km$^2$ per decade, respectively); and 2000–2019 (SIA trends in March and September are $-542 \times 10^3$ km$^2$ per decade and $-1063 \times 10^3$ km$^2$ per decade, respectively).

Fifteen regions in the Arctic Ocean and marginal seas were selected for the regional analysis. The boundaries of these regions on 0.25° × 0.25° lat/lon grid (as in ERA5 data) are shown in Figure 1b. They basically correspond to the broadly accepted geographical definitions of the Arctic seas [67]. In addition, we also considered the inner Arctic Ocean dividing it into two parts, south of 80° N and north of 80° N. The SIA data for all studied Arctic seas calculated in this study is available through Mendeley Data [68].

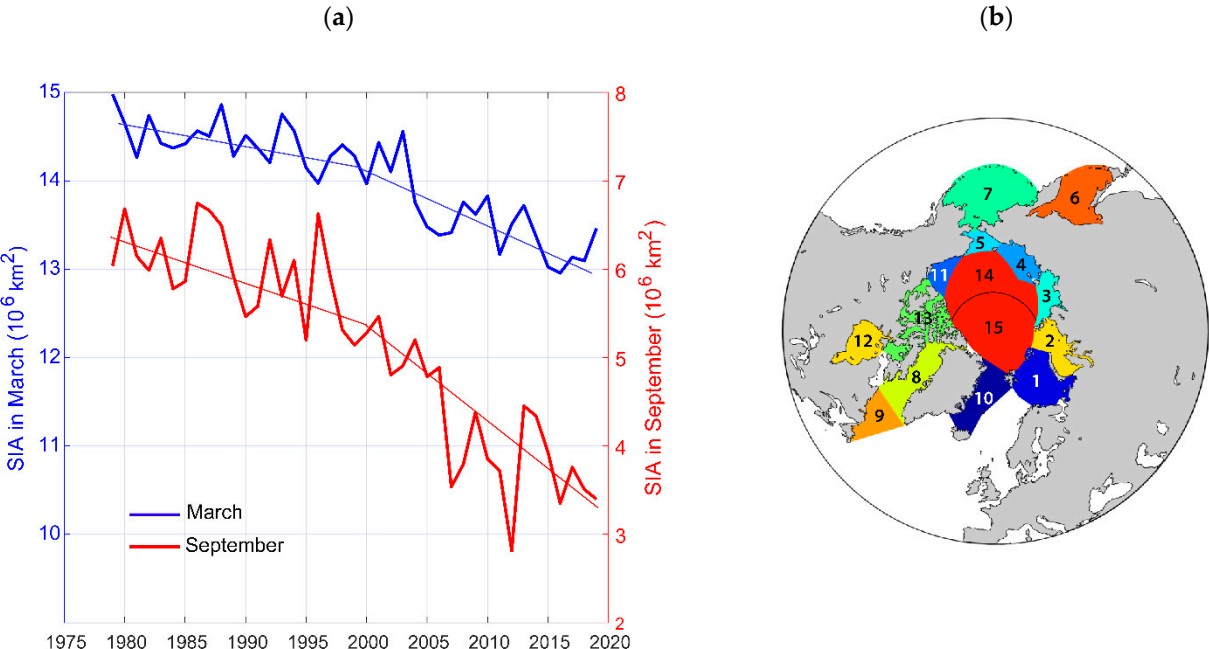

**Figure 1.** (**a**) Northern hemisphere SIA time series from 1979 to 2019 (based on ERA5 SIC data) for March (blue) and September (red) with linear trends for 1979–1999 and 2000–2019 periods. (**b**) The boundaries of the Arctic seas: 1—Barents; 2—Kara; 3—Laptev; 4—East Siberian; 5—Chukchi; 6—Okhotsk; 7—Bering; 8—Baffin; 9—Labrador; 10—Greenland; 11—Beaufort; 12—Hudson Bay; 13—Northwestern Passages; 14—Arctic Ocean (south of 80° N); 15—Arctic Ocean (north of 80° N).

## 3. Results

### 3.1. Seasonal Cycle and Coverage

The NH SIA and SIA in all the studied Arctic regions has a seasonal cycle with a maximum in March and a minimum in August–September. However, it is characterized by noticeable regional differences (Figure 2). In some seas, seasonal sea ice melt begins in April–May (e.g., in Barents, Okhotsk, Bering Seas), and in the most part of the Arctic region, the major sea ice melt occurs later, in May–June. After the minimum in September (when several seas even remain ice-free), the sea ice refreezes rapidly during the two subsequent months in the Laptev, East Siberian, Beaufort Seas, Northwestern Passages, and Arctic Ocean, whereas the other regions do not refreeze until December or even January–February.

However, the seasonal cycle has significantly changed during the satellite era of observation. The NH SIA in 2000–2019, on average, is about $0.88 \times 10^6$ km$^2$ less in the period from December to March and about $1.84 \times 10^6$ km$^2$ less in July–October than in 1979–1999 (grey bars in Figure 2). In general, the amplitude of the seasonal cycle has increased in the recent two decades in comparison to the 1979–1999 period (red and blue curves in Figure 2 for coverage in the two periods, and grey bars for the difference between them). In December–February (DJF), the largest changes in the ice coverage occurred in the Barents Sea. The SIA decreased by 17% in 2000–2019 compared with 1979–1999, and the difference in SIA between these two periods is $280 \times 10^3$ km$^2$. In summer and early autumn, SIA decreased in the Kara Sea by 17% ($294 \times 10^3$ km$^2$) and the Laptev Sea by 19% ($105 \times 10^3$ km$^2$), with almost unchanged coverage in winter (these Seas used to be completely covered with sea ice in winter, but in the recent years, the Kara Sea was just partly sea ice covered [12]). It decreased by 21% in the East Siberian Sea ($154 \times 10^3$ km$^2$), and by 18% ($89 \times 10^3$ km$^2$) in the Beaufort Sea. The greatest changes in the amplitude of the seasonal cycle were observed in the Arctic Ocean south of 80° N, where SIA coverage reduced in July–September by 28% ($532 \times 10^3$ km$^2$) and in October by 24% ($466 \times 10^3$ km$^2$). In the northern part of the Arctic Ocean (north of 80° N) the decline of SIA was smaller in

summer and autumn, only 4%. In the other seas, there were no significant changes in the SIA seasonal cycle.

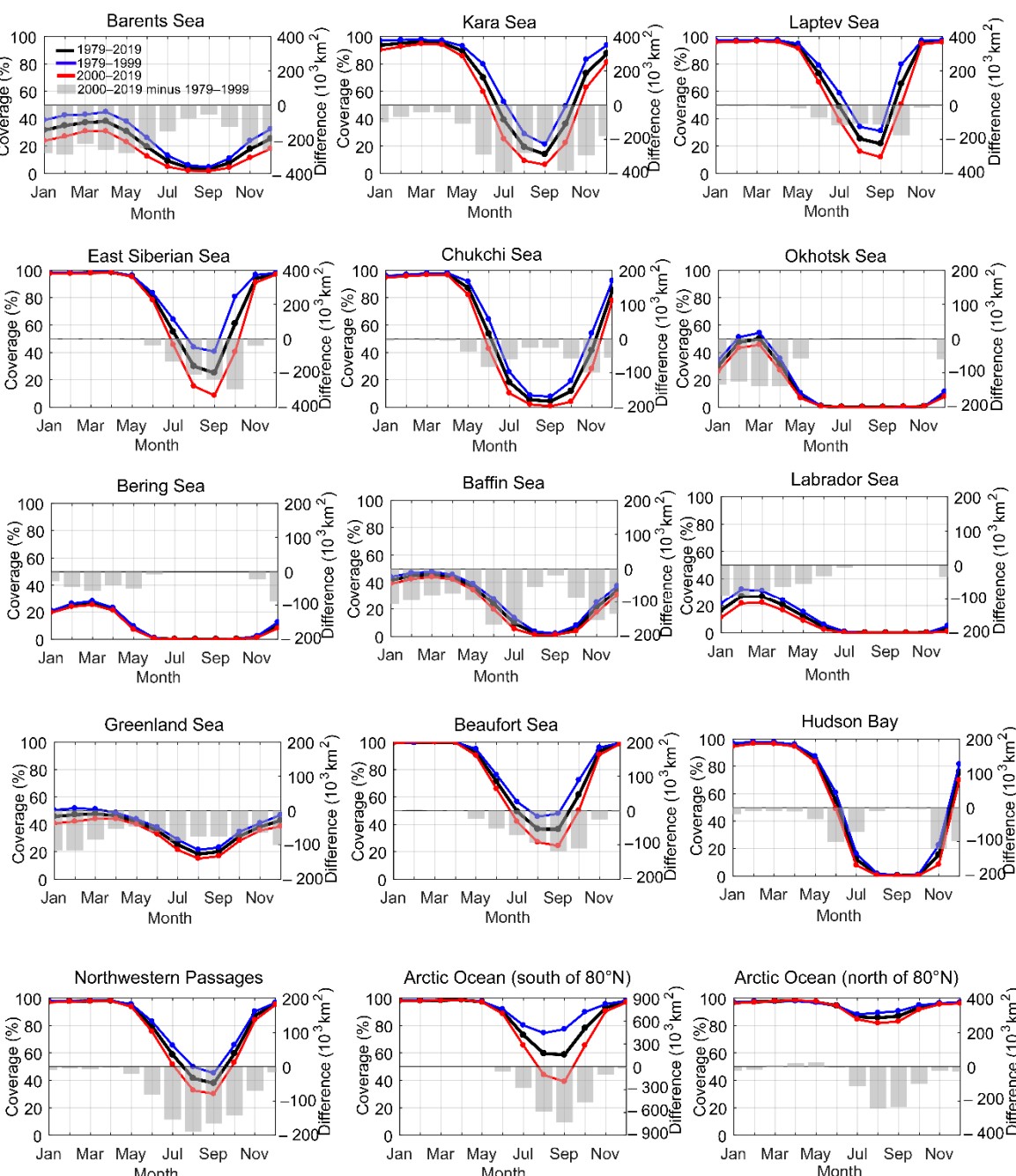

**Figure 2.** Monthly sea ice coverage (% of the selected sea/region area) for individual seas in three periods: 1979–2019 (black curve), 1979–1999 (blue curve) and 2000–2019 (red curve) and differences between mean SIA in 2000–2019 and 1979–1999 (grey bars).

### 3.2. Contributions of Different Seas to the Total Arctic SIA Trend

The contributions of different seas to the total Arctic SIA trend in 1979–2019 are shown in Figure 3. Note that the majority of SIA trends in the Artic seas are negative, i.e., SIA is decreasing. The largest negative trend for total NH SIA is observed in August (864 × 10$^3$ km$^2$ per decade), September (881 × 10$^3$ km$^2$ per decade), and October (859 × 10$^3$ km$^2$/decade). From December to April, the main contribution to the trend is the Barents Sea (33% of the total trend) and the Sea of Okhotsk (24%, where the main

decline occurred before the 21st century). In May, the Barents Sea plays the major role in the negative trend (about 40% of total Arctic SIA trend). The strong negative trend in the Barents Sea has been noted in existing literature (e.g., [69,70]). The large contributions of the Barents Sea and Okhotsk Sea in winter and early spring have also been reported in recent studies (e.g., [60]). The variability of SIA in these seas is more pronounced in winter, whereas in summer they are practically ice-free, and their areas are relatively large; therefore, observed warming air temperature and sea surface temperature (SST), increased Atlantic inflow to the Barents Sea [32,71,72] and changes of atmospheric circulation (in particular, variations in the position of the Aleutian Low) affect SIA fluctuations in the Okhotsk Sea [73], resulting in a strengthening of the decline rate of SIA in these seas and their greater contribution to the decrease of SIA of the total Arctic. In June–July, major contributions to the total trend are made by the Kara (16%) and Barents (20%) Seas, and the Northwestern Passages (10%).

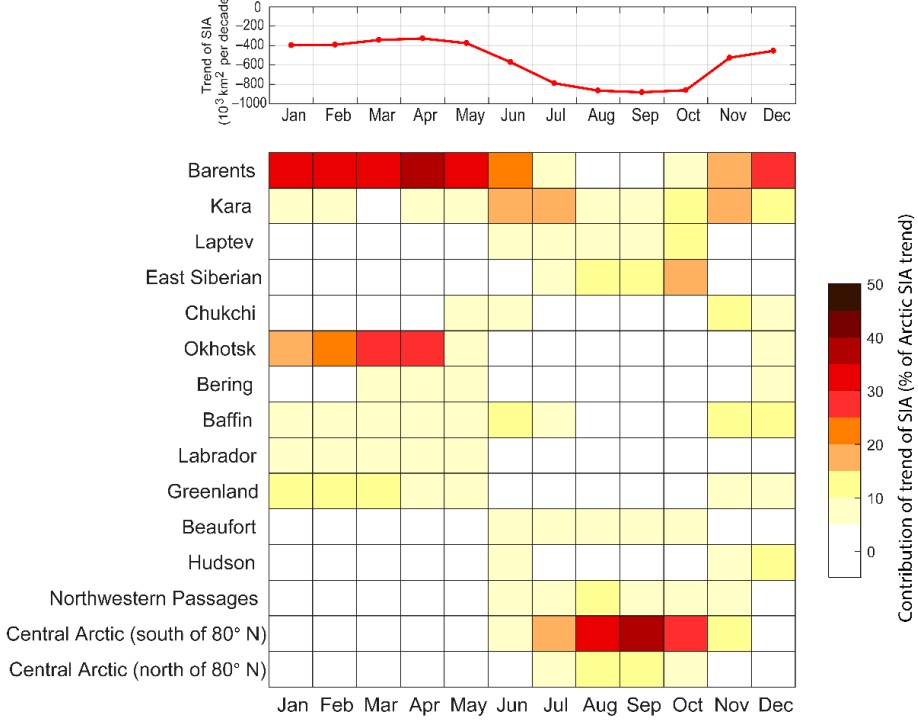

**Figure 3.** Northern hemisphere SIA trends in 1979–2019 for different months (**top**) and the contribution of trends in individual Arctic regions to total northern hemisphere SIA trend (**bottom**) (Contributions exceeding 5% are shaded).

In August–September, the largest part of the total trend is due to sea ice loss in the Arctic Ocean south of 80° N (about 35% of the total trend), north of 80° N (13%), and by the East Siberian Sea (12%). These regions are typically completely ice covered in winter, but in summer and earlier autumn they have a significant variability in SIA and play a major role in the total Arctic trend. Warmer air temperatures and increased SST lead to earlier ice melt onset and enhancement of the positive ice-albedo feedback, which causes a thinning of the sea ice and, as a result, a delay in autumn freeze-up. These features were also highlighted in [60]. In October, when the sea ice begins to refreeze and SIA increases in several regions, the Arctic Ocean to the south of 80° N (26%) and Kara (13%), Laptev (10%), and East Siberian (15%) Seas are the most influential to the total trend. In November–December, the Barents Sea again makes the greatest contribution (23%), as well as the Kara (14%) and Baffin (11%) Seas, and Hudson Bay (10%).

### 3.3. SIA Trends in Different Seas

Monthly SIA trends in different regions for three periods, 1979–1999, 2000–2019 and 1979–2019, are shown in Figure 4. The structure of SIA trends during the whole period 1979–2019 (Figure 4a) is similar to the structure of sea ice extent trends for different seas shown by [60]. However, the changes in SIA during 1979–2019 were not linear, and the structure of trends for the whole period 1979–2019 differs noticeably from that in 1979–1999 (Figure 4b), appearing more similar to the trends for 2000–2019 (Figure 4c). The trends in absolute values are also different.

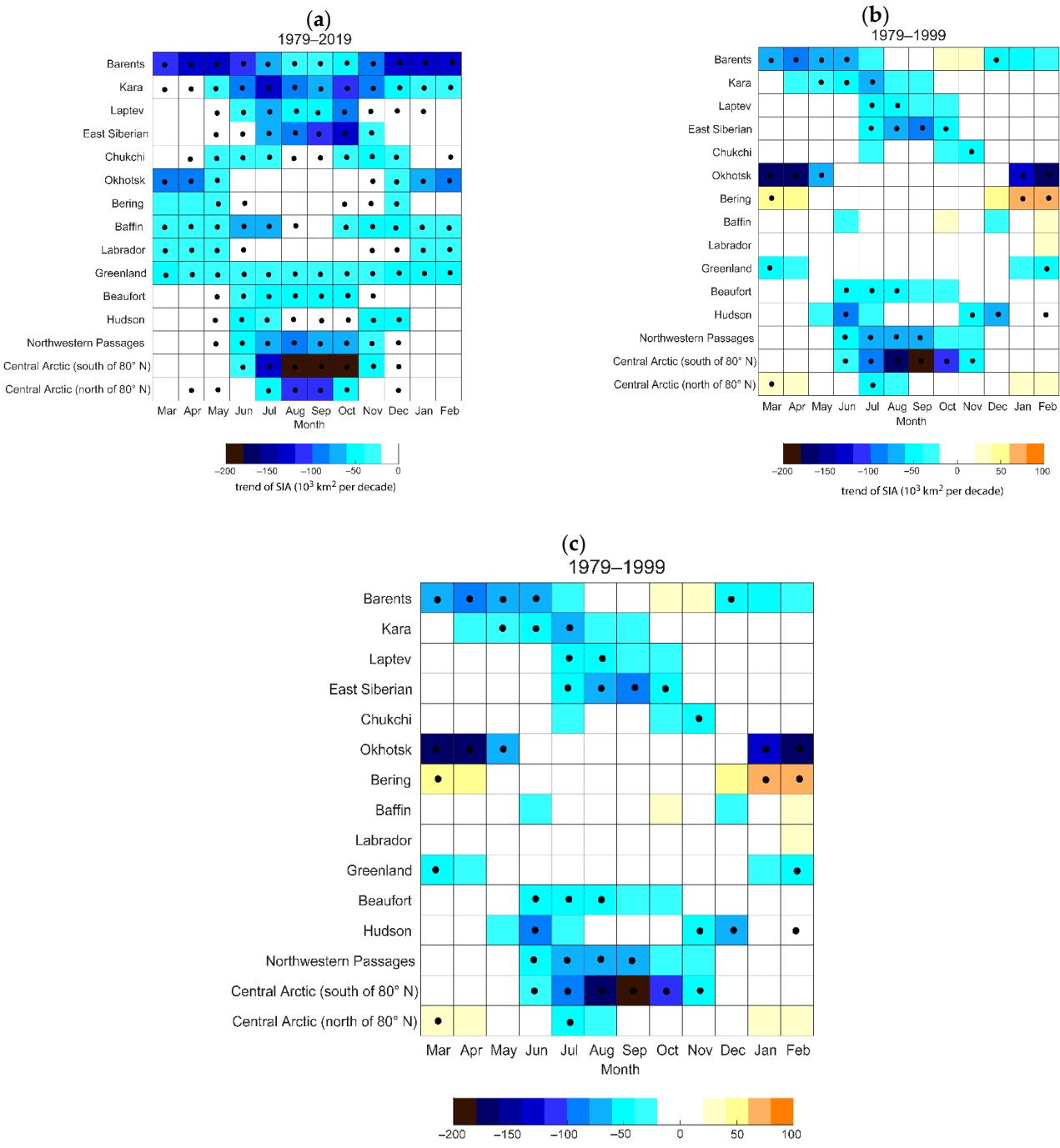

**Figure 4.** Trends for monthly SIA in (**a**) 1979–2019, (**b**) 1979–1999, and (**c**) 2000–2019. (Statistically significant (95% confidence level) trends are marked with dots, according Mann–Kendall Test).

In comparison with 1979–2019, the 2000–2019 decline in SIA increases mostly from June to November in the Kara, Laptev, East Siberian, Beaufort Seas, the Northwestern Passages and Central Arctic (both south and north of 80° N). In June–July, the largest difference in SIA trend between the 2000–2019 and 1979–1999 periods is observed in the Laptev ($52 \times 10^3$ km$^2$ per decade), Beaufort ($41 \times 10^3$ km$^2$ per decade) Seas, and in the Arctic Ocean south of 80° N ($96 \times 10^3$ km$^2$ per decade). Here, the rate of reduction of SIA in July almost more than doubled over the past 20 years (from $75 \times 10^3$ to $154 \times 10^3$ km$^2$ per decade). The negative trend in July also increased in the Northwestern Passages (by $43 \times 10^3$ km$^2$ per decade). In August–September, the reduction of SIA notably accelerated in the Arctic Ocean south of 80° N by $201 \times 10^3$ km$^2$ per decade, and north of 80° N by $186 \times 10^3$ km$^2$ per decade. In the same months, the trend in the Kara Sea changed from near zero in 1979–1999 to $91 \times 10^3$ km$^2$ per decade in 2000–2019, caused, in general, by increased Atlantic inflow to this region. The most significant changes are observed in October, when the rate of NH SIA decline increased by $0.97 \times 10^6$ km$^2$ per decade in 2000–2019 compared with 1979–1999. The strongest trend changes occurred in the Arctic Ocean south of 80° N (by $231 \times 10^3$ km$^2$ per decade). In the northern part of the Arctic Ocean around the Pole, the decreasing SIA trend has also strengthened (by $106 \times 10^3$ km$^2$ per decade). The large changes in SIA trend in October between the two periods are also observed in the Kara (by $186 \times 10^3$ km$^2$ per decade), Laptev (by $131 \times 10^3$ km$^2$ per decade), and East Siberian (by $124 \times 10^3$ km$^2$/decade) Seas. These changes are mainly caused by the increasing air temperature and SST in the polar region. Accelerated warming trends were observed over most of the Arctic in the recent decades (e.g., [9,74,75]) accompanied by a reduction of multi-year sea ice [76].

From December to March, SIA melting rates increased in the Barents, and Greenland Seas. In the Bering Sea and Central Arctic north of 80° N, SIA trends even changed from positive values in 1979–1999 to strongly negative rates in 2000–2019. In the Bering Sea and Central Arctic, the open-water period lengthened and freeze-up dates in autumn were delayed. In the Barents Sea, the difference in SIA trends between 2000–2019 and 1979–1999 is $116 \times 10^3$ km$^2$/decade (averaged for December to March). The variations in the Atlantic waters inflow are a dominant influence on the recently accelerated SIA loss in the Barents Sea [2,32,72]. This process leads to an increased accumulation of heat in the upper ocean layer during the summer. This additional heat favors more active winter thermohaline convection and transport of heat from the Atlantic water layer to the sea ice. This causes the sea ice thinning and longer open-water season, which, in turn, leads to an increased accumulation of heat in the ocean in the following summer [70,77]. In April–May, the SIA trends notably decrease in 2000–2019 in the Barents Sea by $72 \times 10^3$ km$^2$ per decade, and in the Bering Sea by $116 \times 10^3$ km$^2$ per decade.

### 3.4. Spatial Changes of SIC Trends

The SIC trends also significantly changed in 2000–2019 compared with 1979–1999. In March, the area of intense sea ice reduction in the Greenland Sea unexpectedly moved southward in 2000–2019, and extended towards the northeastern, eastern, and southeastern margins of the Barents Sea (Figure 5). An area of significant sea ice loss appeared to the east and to the north of Svalbard. In addition, a region with strong sea ice loss emerged in the southern part of the Bering Sea and in the southwestern margins of Okhotsk Sea (not shown).

In September, the area with noticeable SIC loss in 2000–2019 covers a larger region and extends to the north of the Franz Josef Land, the northern boundaries of the Kara, East Siberian, Laptev, and Beaufort Seas, and moves into the inner Arctic Ocean. More areas with significant decrease appear in the Northwestern Passages.

The area with trends exceeding 10% per decade in September grew from 1.48 million to 2.72 million km$^2$ in March and from 1.86 million to 3.69 million km$^2$ for 1979–1999 and 2000–2019 periods, respectively.

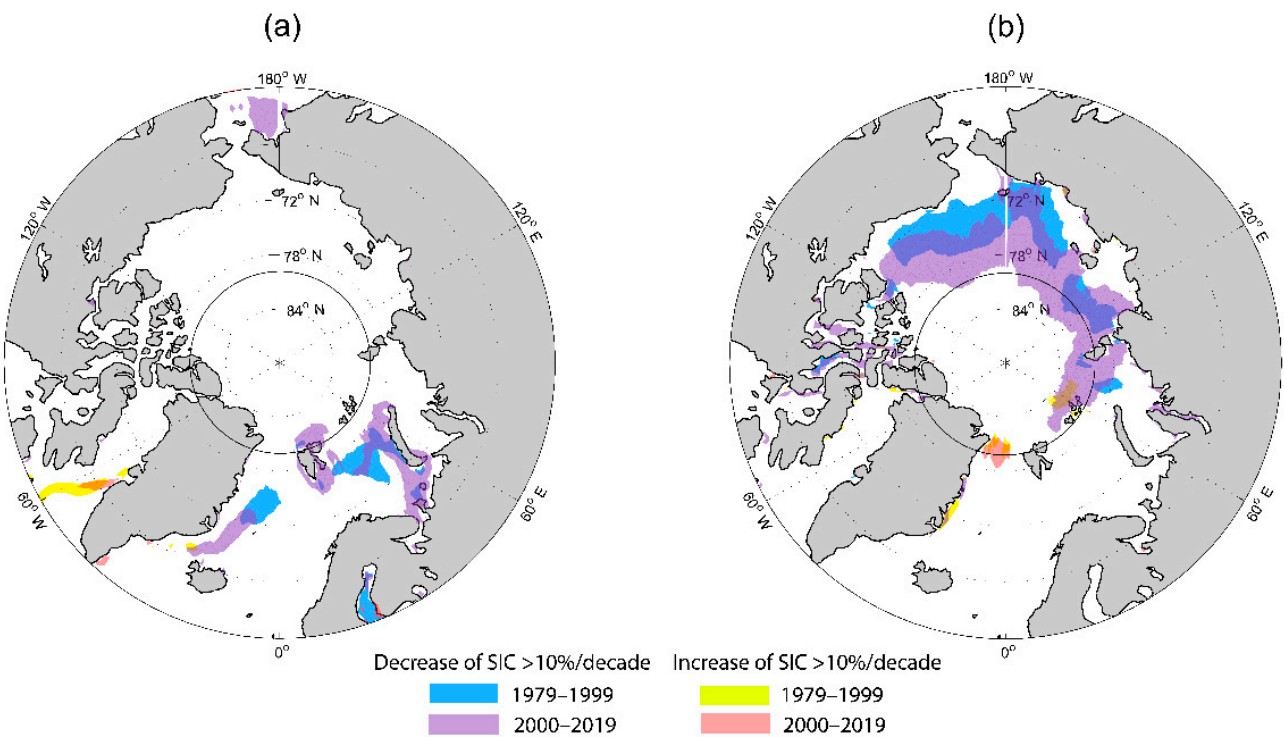

**Figure 5.** Areas of noticeable SIC trends (more than 10% per decade) in 1979–1999 and 2000–2019 for (**a**) March and (**b**) September (Black circle shows 80° N latitude). All indicated (shaded) trends are statistically significant ($p < 0.1$).

### 4. Discussion

Disentangling the major factors accelerating the sea ice reduction is a complicated task. Here, we would like to initiate and support further investigations of this problem by comparing SIA trends as observed and simulated by the latest generation of CMIP6 climate models. Trends for the 1979–1999 and 2000–2019 periods for September and March are shown in Figure 6. A majority of models do exhibit a faster September SIA decline in the recent period, whereas in March, more than a half of the models show a stronger decline in the end of the 20th century. For several models, the SIA decline was faster in 1979–1999 than in 2000–2019 (12 models in March and 7 models in September). In March, 7 models even demonstrate positive trends in 2000–2019. The large uncertainties can be caused by different representations of SIA-related processes in the models and great deviation of SIA within even a particular model, with some models showing an earlier start of fast SIA decline, than others. This suggests an important role of internal climate variability in winter in driving decadal sea ice trends in the Arctic (e.g., [21,33]). We also note that Figure 6 shows Arctic wide SIA changes in CMIP6 models without breaking it up regionally, compared with the ERA5 data used in this study. Regional changes can be another source of error and additional uncertainties.

The internal climate variability that could be a major cause of the described differences in SIA characteristics between the 1979–1999 and 2000–2019 periods is also suggested as a reason for a sharp increase in Arctic amplification in 1999 recently found by Chylek et al. [78] based on comparison of observed and simulated (CMIP6) temperature changes.

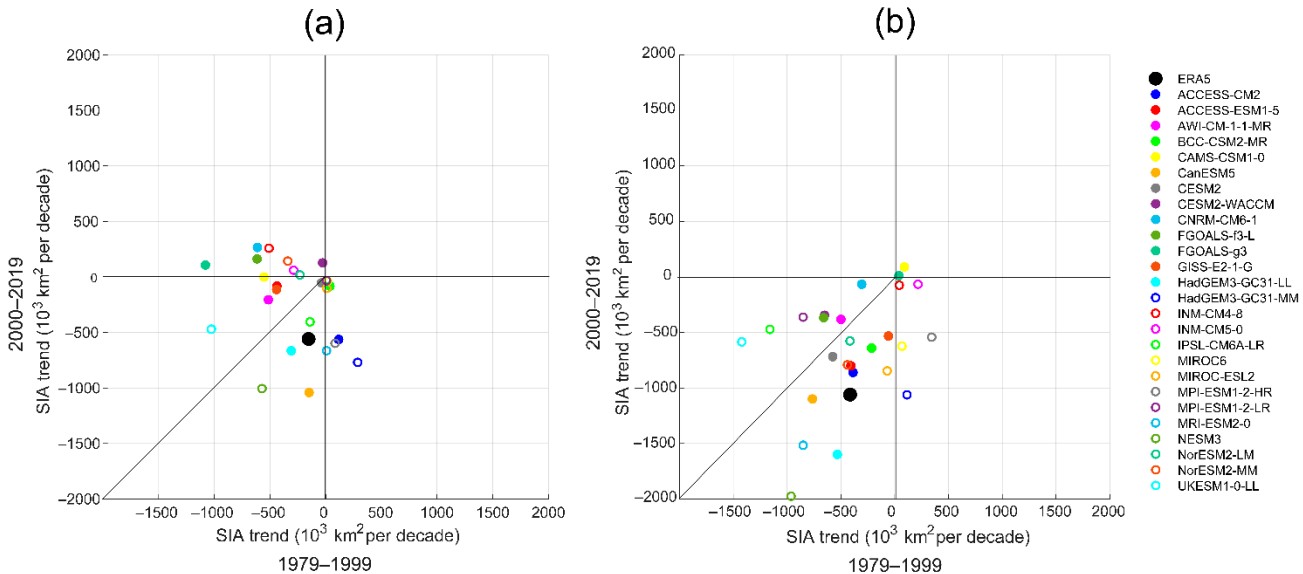

**Figure 6.** Relationship between Arctic SIA trends in 1979–1999 and 2000–2019 for 26 CMIP6 models and ERA5 data in (**a**) March and (**b**) September.

## 5. Conclusions

The contributions of SIA trends in the Arctic seas to the total NH SIA trend during the 1979–2019 period were estimated. From December to April, the largest contributions to the NH SIA changes were from sea ice retreat in the Barents Sea and the Sea of Okhotsk. In May, the major contribution in the total SIA trend was made by SIA changes in the Barents Sea. In June–July, the Kara, Laptev, East Siberian Seas and the Northwestern Passages play the most important role in the total NH trend. In August–September, the SIA decline in the inner Arctic Ocean south of 80° N and in the East Siberian Sea play the major role in the total NH SIA trend. In October, the Arctic Ocean to the south of 80° N, the Kara, Laptev and East Siberian Seas contribute most to the total Arctic SIA trend. In November–December, the Barents Sea makes the greatest contribution, as well as the Kara, Baffin Seas, and Hudson Bay.

In contrast to the previous studies (e.g., [12,59,60]) that focused of sea ice changes in the whole period from 1979 onwards, in this study we compared regional changes between the period of the modern accelerated decline and the previous, more gradual decline. In the Barents Sea, the mean winter SIA trends in 2000–2019 increased more than two-fold in comparison with the trends in 1979–1999. In April–May, the SIA trends notably decrease in 2000–2019 in the Barents Sea and in the Bering Sea. In June–July, SIA reduction in the 21st century accelerated most strongly in the Laptev, Beaufort Seas, in the inner Arctic Ocean south of 80° N, and in the Northwestern Passages. In August–September, the reduction of SIA accelerated notably in the Arctic Ocean (both south and north of 80° N), and in the Kara Sea. The strongest changes in trends during 2000–2019 compared with the 1979–1999 period occurred in October in the Arctic Ocean south of 80° N. In the northern part of the Arctic Ocean around the Pole, in the Kara, Laptev, and East Siberian Seas, the decreasing trend has also strengthened in October.

It was shown that the amplitude of the SIA seasonal cycle increased in 2000–2019 in comparison with the 1979–1999 period. The largest changes in the amplitude between the 2000–2019 and 1979–1999 periods occurred in the Barents, Kara, Laptev, East Siberian, and Beaufort Seas. The most dramatic changes in SIA were observed in the Arctic Ocean south of 80° N, where the SIA coverage reduced by almost half in the summer and early autumn. When considering mean SIA changes between the two chosen periods, the largest changes in winter occurred in the Barents Sea. In summer and early autumn, SIA strongly decreased in the Kara, Laptev, East Siberian and Beaufort Seas.

Spatial patterns of SIC trends for the 1979–1999 and 2000–2019 periods in March and September were presented. In summer, the area of sea ice concentration trends exceeding 10%/decade increased from 1.87 to 3.69 million km$^2$, extending northward around the pole except for the Canadian Archipelago and Greenland sectors of the Arctic Ocean. In winter, changes in the areas of SIA decrease are smaller (from 1.48 to 2.72 million km$^2$) and mostly located in the Atlantic sector spreading to the eastern part of the Barents Sea and east of Svalbard in the beginning of the 21st century.

The accelerated Arctic sea ice melt in the beginning of the 21st century can be related to the increasing anthropogenic greenhouse forcing, and also may reflect a transition to a new dynamic state with increasing ocean and atmosphere heat transport to the Arctic [30,32,45] with triggering positive feedbacks in the Arctic climate system (e.g., [4,77]), thus implying a possible tipping point [53].

**Author Contributions:** Conceptualization, T.A.M. and V.A.S.; methodology, T.A.M. and V.A.S.; software, T.A.M.; validation, T.A.M.; formal analysis, T.A.M. and V.A.S.; investigation, T.A.M.; writing—original draft preparation, T.A.M. and V.A.S.; writing—review and editing, T.A.M. and V.A.S.; visualization, T.A.M.; supervision, V.A.S. All authors have read and agreed to the published version of the manuscript.

**Funding:** This work was supported by the Russian Ministry of Science and Education (Agreement No. 075-15-2020-776).

**Institutional Review Board Statement:** Not applicable.

**Informed Consent Statement:** Not applicable.

**Data Availability Statement:** General data used in this study is archived in the repository (Datasets Generated: "Mendeley Data": Sea ice area in different Arctic Seas (timeseries based on ERA5), Mendeley Data, V1, https://doi.org/10.17632/v8xt3xy87t.1 [accessed date: 19 April 2022].

**Conflicts of Interest:** The authors declare no conflict of interest.

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
