# Peer review of "Regional Features of the Arctic Sea Ice Area Changes in 2000–2019 versus 1979–1999 Periods"

_atmosphere, doi:10.3390/atmos13091434_

Round 1
Reviewer 1 Report
The manuscript "Regional features of the Arctic sea ice area changes in 2000-2019 versus 1979-1999 periods" by Matveeva and Semenov is a high quality research work and a carefully prepared presentation. I recommend only a minor revision before its publication.
The most important point is the need to justify the choice of periods. Ad hoc explanation is given but not good enough. There are many objective methods to identify structural break points in time series, e.g., Werner, R., Valev, D., Danov, D., & Guineva, V. (2015). Study of structural break points in global and hemispheric temperature series by piecewise regression. Advances in Space Research, 56(11), 2323–2334. https://doi.org/10.1016/j.asr.2015.09.007. It would be nice to see use of them for the periods' justification.
There are regular misprints of numbers like 103 instead of 103.
It would be perhaps beneficial to add to the discussion anther conceptual view on the trend breaking, e.g., Chylek, P., Folland, C., Klett, J. D., Wang, M., Hengartner, N., Lesins, G., & Dubey, M. K. (2022). Annual Mean Arctic Amplification 1970–2020: Observed and Simulated by CMIP6 Climate Models. Geophysical Research Letters, 49(13). https://doi.org/10.1029/2022GL099371
Reviewer 2 Report
Please find attached the manuscript with the comments made directly on it. I suggest minor revisions.

Author Response
We are grateful to the Reviewer for scrutinizing our paper and for constructive comments that helped us to improve the manuscript. We have incorporated the suggested changes and corrected a number of misprints and style errors caught by the Reviewer into the revised manuscript. All revisions to the manuscript were marked in a “track changes” mode.
The Figure 2 was corrected as suggested. The y-axis for the difference between mean SIA in 2000‒2019 and 1979‒1999 now contains values from -400 to 400, except for the Arctic Ocean south of 80°N where the changes of SIA are too large and do not fit into this range.
The sentence “The sea ice in the Northwest Passages declined rapidly due to decrease of sea ice thickness linked to warmer SST and salinity” in sub-section 3.3. SIA trends in different Seas was deleted.
We added the suggested comment about CMIP6 models in the Discussion.